# Tissue-resident natural killer (NK) cells are cell lineages distinct from thymic and conventional splenic NK cells

Dorothy K Sojka[1], Beatrice Plougastel-Douglas[1], Liping Yang[1], Melissa A Pak-Wittel[1], Maxim N Artyomov[2], Yulia Ivanova[2], Chao Zhong[3], Julie M Chase[1], Paul B Rothman[4], Jenny Yu[1], Joan K Riley[5], Jinfang Zhu[3], Zhigang Tian[6,7], Wayne M Yokoyama[1,8]*

[1]Rheumatology Division, Washington University School of Medicine, St. Louis, United States; [2]Department of Pathology and Immunology, Washington University School of Medicine, St. Louis, United States; [3]Molecular and Cellular Immunoregulation Unit, Laboratory of Immunology, National Institute of Allergy and Infectious Diseases, National Institutes of Health, Bethesda, United States; [4]The Johns Hopkins University School of Medicine, Baltimore, United States; [5]Obstetrics and Gynecology Division, Washington University School of Medicine, St. Louis, United States; [6]Department of Immunology, School of Life Sciences, University of Science and Technology China, Hefei, China; [7]Hefei National Laboratory for Physical Sciences at the Microscale, Hefei, China; [8]Howard Hughes Medical Institute, Washington University School of Medicine, St. Louis, United States

**Abstract** Natural killer (NK) cells belong to the innate immune system; they can control virus infections and developing tumors by cytotoxicity and producing inflammatory cytokines. Most studies of mouse NK cells, however, have focused on conventional NK (cNK) cells in the spleen. Recently, we described two populations of liver NK cells, tissue-resident NK (trNK) cells and those resembling splenic cNK cells. However, their lineage relationship was unclear; trNK cells could be developing cNK cells, related to thymic NK cells, or a lineage distinct from both cNK and thymic NK cells. Herein we used detailed transcriptomic, flow cytometric, and functional analysis and transcription factor-deficient mice to determine that liver trNK cells form a distinct lineage from cNK and thymic NK cells. Taken together with analysis of trNK cells in other tissues, there are at least four distinct lineages of NK cells: cNK, thymic, liver (and skin) trNK, and uterine trNK cells.

*For correspondence:
yokoyama@dom.wustl.edu

**Competing interests:** The authors declare that no competing interests exist.

**Reviewing editor**: Diane Mathis, Harvard Medical School, United States

## Introduction

Immune cells migrate throughout the body where they search tissues for pathological events induced by invading pathogens or emerging tumors, for example. Upon encounter with these events, circulating immune cells stop and respond in collaboration with yet other immune cells, often in organized lymphoid tissues. The subsequent orchestrated host immune response controls the pathological process by directing relevant immune cells to the damaged tissue. In contrast to circulating immune cells are tissue-resident immune cells which already reside in selected organs where they appear to be poised to deliver immune responses. However, how tissue-resident immune cells contribute to host responses is less well understood and will be aided by identifying factors that distinguish circulating and tissue-resident immune cells.

Natural killer (NK) cells are components of the innate immune system (*Yokoyama, 2013*). Initially described on the basis of their inherent capacity to kill tumor cells without prior sensitization, NK cells

**eLife digest** Our immune system has white blood cells that migrate throughout the body in search of invading microbes or diseased and damaged cells. When these events are encountered, the white blood cells move into the affected tissue and launch an immune response to eliminate the threat.

Natural killer cells are white blood cells that kill cells that are infected with viruses or are cancerous. Most of what is known about conventional natural killer cells is derived from studying the spleen, which filters the blood and contains many immune cells. Natural killer cells also circulate around the body or are found within other tissues, and it was thought that both types of cells were either the same, or that one type could develop into the other. However, the thymus—an organ that is another source of white blood cells—contains a sub-population of natural killer cells that are distinct from the conventional splenic natural killer cells. Furthermore, recent work revealed the existence of two types of natural killer cells within the liver: some of these cells were similar to the conventional splenic natural killer cells that circulate throughout the body, while others appeared to be 'tissue-resident' natural killer cells that were poised to deliver an immune response.

Now Sojka et al. show that the tissue-resident natural killer cells found in the liver are a distinct lineage of cells. These cells mature independently from the conventional natural killer cells found in the spleen, and the natural killer cells found in the thymus. Moreover, the skin contains tissue-resident natural killer cells similar to those in the liver; whilst natural killer cells that had previously been discovered in the uterus were shown to contain a fourth distinct tissue-resident lineage.

The work of Sojka et al. will encourage a full re-evaluation of the roles played by natural killer cells to determine which populations of these cells are responsible for implementing immune responses. Furthermore, a more thorough understanding of how tissue-resident natural killer cells function to eliminate diseased or damaged cells, such as cancerous cells, could also contribute to future efforts to develop new anti-cancer treatments.

are now known to participate in a wide variety of immune responses, such as early control of viral infections. In addition, they can respond to pro-inflammatory cytokines by producing yet other inflammatory cytokines, such as interferon-γ (IFNγ), their signature cytokine that can influence adaptive immune cells.

NK cells require IL-15 and its cognate receptor, IL-15R, for development (*Cao et al., 1995*; *DiSanto et al., 1995*; *Suzuki et al., 1997*; *Lodolce et al., 1998*; *Kennedy et al., 2000*). In knockout mice lacking IL-15 or any chain of the trimeric IL-15R (α, β, γ), splenic NK cells are absent. The development of splenic NK cells is thought to occur largely if not exclusively in the bone marrow (BM) where cells committed to the NK cell lineage undergo a series of putative developmental stages, characterized by acquisition and loss of various surface markers, including cytokine receptors, NK cell receptors, and integrins (*Kim et al., 2002*; *Yokoyama et al., 2004*; *Di Santo, 2006*). Out in the periphery, mature splenic NK cells can be further distinguished by differential expression of CD11b and CD27 (*Kim et al., 2002*; *Hayakawa and Smyth, 2006*). Thus, conventional splenic NK cells display developmental markers associated with maturation.

NK cells require certain transcription factors for development (*Hesslein and Lanier, 2011*), in particular NFIL3 (E4BP4), described as the NK cell-specification factor (*Di Santo, 2009*). Mice lacking NFIL3 have essentially no splenic NK cells though other organs were not thoroughly examined (*Gascoyne et al., 2009*; *Kamizono et al., 2009*; *Kashiwada et al., 2010*). The related t-box transcription factors, Tbet (Tbx21) and eomesodermin (Eomes), play more complex roles in NK cell development (*Townsend et al., 2004*; *Gordon et al., 2012*). In the absence of Tbet, splenic NK cells display an immature phenotype, and a subpopulation of NK cells in the liver is absent, consistent with overlapping and cooperative roles of Tbet and Eomes in NK cell development. Alternatively, Tbet may direct the development of a separate lineage of NK cells. Current data cannot definitively distinguish between these possibilities for the role of Tbet and Eomes in NK cell development.

Classically studied in the mouse spleen and expressing the NK1.1 antigen on CD3ε-negative cells, NK cells are also present in solid organs, such as the thymus, uterus, and liver (*Yokoyama, 2013*). Like the conventional NK (cNK) cells in the spleen, thymic NK cells are cytotoxic and require IL-15 but they

differ from cNK cells by their characteristic expression of CD127 (IL-7 receptor α) and requirement for a thymus where they can arise from early thymocyte precursors (*Vosshenrich et al., 2006*; *Ribeiro et al., 2010*; *Vargas et al., 2011*). Furthermore, thymic NK cells uniquely require the transcription factor GATA-3 for development (*Vosshenrich et al., 2006*). NK cells are normally present in the non-pregnant uterus (*Parr et al., 1991*; *Yadi et al., 2008*; *Mallidi et al., 2009*) but have been mostly studied after they expand at the site of embryo implantation during pregnancy (*Moffett and Loke, 2006*; *Hatta et al., 2012*). Like cNK cells, uterine NK (uNK) cells require IL-15 for development (*Ashkar et al., 2003*). In addition, they are cytotoxic as they express perforin and granzymes, and they produce IFNγ (*Parr et al., 1990*; *Ashkar et al., 2000*). Interestingly, however, uNK cells appear relatively normal in Tbet-deficient mice (*Tayade et al., 2005*) and recent studies suggest that a subset of uNK cells can be distinguished from cNK cells (*Yadi et al., 2008*). Thus, NK cell subsets can be identified in different tissues that appear to be distinguishable from cNK cells.

In the liver, we recently showed that there are two populations of NK cells, distinguished by mutually exclusive expression of CD49a and DX5 (*Peng et al., 2013*). Phenotypically, CD49a$^-$DX5$^+$ are very similar to splenic cNK cells whereas CD49a$^+$ DX5$^-$ are unlike splenic cNK cells. In parabiotic mice, the host liver contains CD49a$^+$ DX5$^-$ NK cells of host origin and circulating CD49a$^-$DX5$^+$ NK cells derived from both host and the other parabiont, indicating that the CD49a$^+$ DX5$^-$ cells are tissue-resident NK (trNK) cells whereas the CD49a$^-$DX5$^+$ cells are cNK cells. The trNK cells appear similar to immature cNK cells because they express similar markers (NK1.1, NKp46) but low levels of CD11b, are DX5$^-$, and display high levels of TNF-related apoptosis-inducing ligand (TRAIL) (*Kim et al., 2002*; *Di Santo , 2006*; *Peng et al., 2013*). However, the liver trNK cells appear to be distinct from immature BM cNK cells because they are cytotoxic and express CD49a. Nonetheless, the liver trNK cells could represent an intermediate cell stage in the development of mature splenic cNK cells from NK cell precursors.

Consistent with this possible developmental intermediate stage, there is a complete absence of TRAIL$^+$ NK cells in the livers of Tbet-deficient mice (*Gordon et al., 2012*). Importantly, this analysis was done before the CD49a$^+$ DX5$^-$ trNK cells in the liver were described. Therefore, while the subset of TRAIL$^+$ liver NK cells lacking in Tbet-deficient mice could represent a failure of maturation of cNK cells, an alternative interpretation of these results is that there may be two different NK cell lineages, one of which is completely Tbet-dependent.

The current data therefore suggest that there are three possible origins of the liver trNK cells. (1) They could be related to thymic NK cells because GATA-3-deficient NK cells are defective in homing to the liver (*Samson et al., 2003*). (2) They could represent an intermediate stage in the development of NK cell precursors into mature splenic NK cells. (3) They could represent an alternative NK cell lineage, distinct from both cNK and thymic NK cells. If these relationships can be resolved, it will then be important to determine how liver trNK cells are related to other trNK cells, such as uNK cells, and to the expanding list of innate lymphoid cells (ILCs) (*Spits and Cupedo, 2012*; *Spits et al., 2013*).

Herein we examined the liver trNK cells in detail with respect to thymic NK, splenic cNK, and trNK cells in other organs. Based on phenotypic characteristics and transcription factor requirements, our studies support the presence of at least four populations of NK cells: cNK (spleen and circulating), thymic, liver (and skin) trNK cells, and uNK cells. Furthermore, these studies indicate that trNK cells can be distinguished from each other and from ILCs.

## Results

### CD49a$^+$DX5$^-$ trNK cells in the liver are distinct from CD127$^+$ thymic NK cells

Thymic NK cells require GATA-3 (*Vosshenrich et al., 2006*), and GATA-3-deficient NK cells have a defect in liver migration and IFNγ production (*Samson et al., 2003*), raising the possibility that the liver trNK cells may be related to thymic CD127$^+$ NK cells. However, as compared to thymic NK cells, liver trNK (CD49a$^+$ DX5$^-$) cells did not clearly express the prototypic thymic NK cell marker, CD127 (*Figure 1A*). To more definitively determine the relationship between these NK cell subsets, we examined athymic nude mice (*Foxn1$^{-/-}$*) which lack thymic NK cells but possess liver trNK cells at a somewhat higher percentage when compared to wildtype (WT) controls (*Figure 1B,C*). Also, liver trNK cell numbers are preserved or higher in nude mice (*Figure 1D*). In addition, we examined mice with

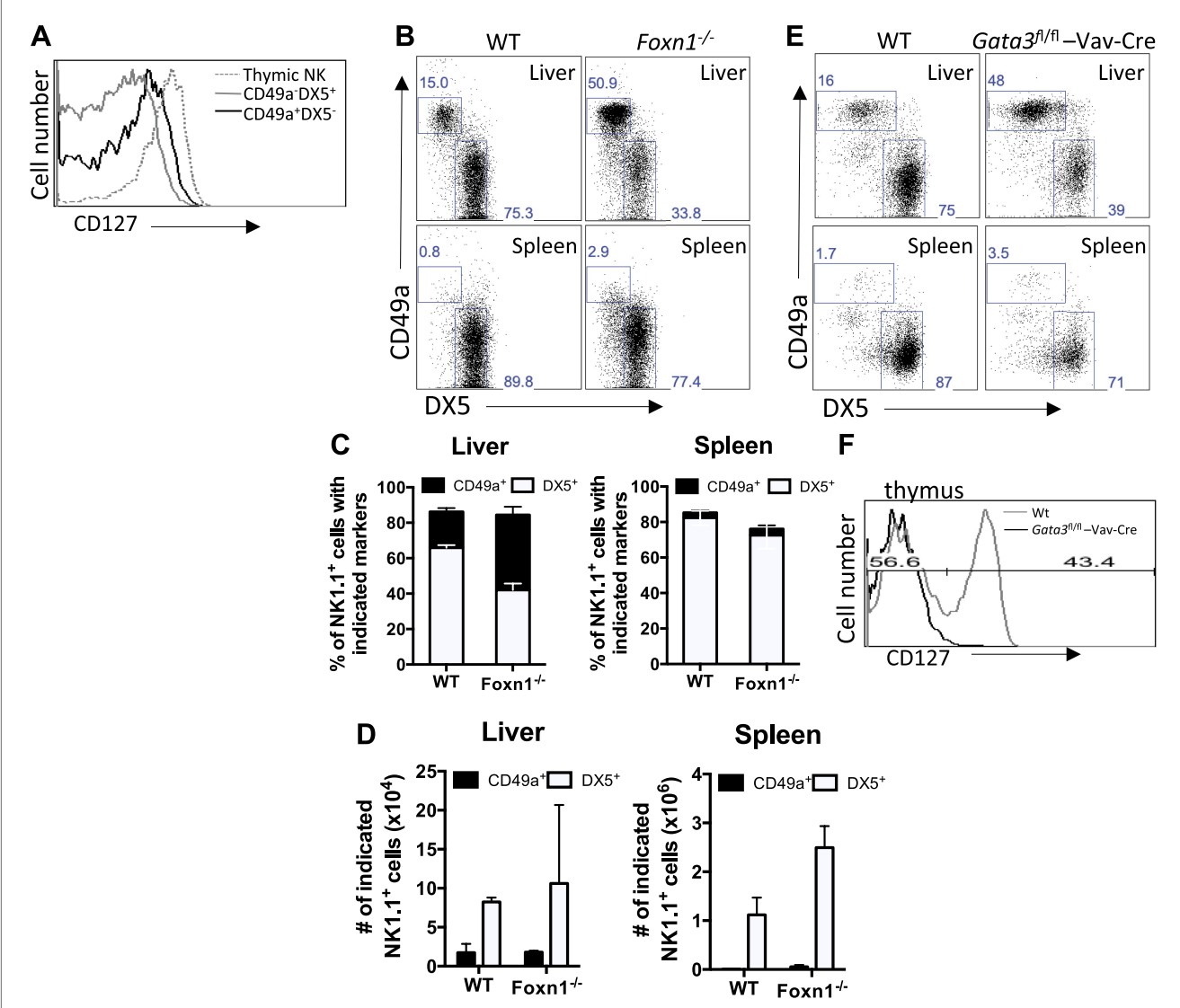

**Figure 1**. CD49a+ DX5− trNK cells in the liver are distinct from the CD127+ thymic NK cells. (**A**) CD127 is poorly expressed on liver trNK cells. Thymi and livers were isolated from WT C57BL/6NCr mice, stained, and flow cytometry was performed. The histogram displays the expression level of CD127 on thymic NK cells and CD49a+DX5− and CD49a−DX5+ NK cells in the liver. Gated on live CD3−CD19−NK1.1+ cells. (**B–D**) Liver trNK cells are present in nude mice. Spleens and livers were isolated from WT C57BL/6NCr and *Foxn1−/−* mice, stained, and flow cytometry performed. The dot plots (**B**) were gated on live CD3−CD19−NK1.1+ cells and display the percentage expressing CD49a and DX5 in each mouse strain in the liver and spleen, as indicated. Stacked bar graphs represent the percentage (**C**) and total number (**D**) of CD3−CD19−NK1.1+ cells that express CD49a and DX5 in the liver and spleen of the WT and *Foxn1−/−* mice. Experiments were performed three independent times. (**E**) Liver trNK cells are present in GATA-3 conditional-deficient mice. Spleens and livers were isolated from WT and *Gata3fl/fl*-Vav-Cre mice, stained, and flow cytometry was performed. The dot plots were gated on live CD3−CD19−NK1.1+ cells and display the percentage expressing CD49a and DX5 in each mouse strain in the liver (top panels) and the spleen (bottom panels). Dot plots represent one of two independent experiments performed. (**F**) Thymic NK cells are absent in *Gata3fl/fl*-Vav-Cre mice. Cells from the thymi of WT and *Gata3fl/fl*-Vav-Cre mice were isolated, stained, and flow cytometry performed. The histogram displays the expression level of CD127 on thymic NK cells gated on live CD3−CD19−NK1.1+ cells. Histogram represents one of two independent experiments performed.

hematopoietic cell deficiency of GATA-3 due to expression of Cre under control of Vav which is expressed in all hematopoietic cells. The liver trNK cells from *Gata3fl/fl*-Vav-Cre mice were present (*Figure 1E*) whereas CD127+ thymic NK cells were absent in these mice (*Figure 1F*), as previously shown (*Vosshenrich et al., 2006*). Taken together, these data indicate that the trNK cells from the liver develop independent of the thymus or GATA-3, and can be clearly distinguished from thymic NK cells.

## Distinct lineages of NK cells suggested by RNA-seq analysis

To further define the molecular similarities and differences between trNK cells in the liver, and circulating liver and splenic cNK cells, we performed RNA deep sequencing (RNA-seq) of sorted NK cells from *Rag1*$^{-/-}$ mice. Basic hierarchical clustering of the three populations revealed that liver trNK cells are distinct from the liver and splenic cNK cells, which in turn, are more closely related to each other than to liver trNK cells (*Figure 2A*). (A higher resolution figure displaying genes most differentially expressed on liver trNK cells is shown in *Figure 2—figure supplement 1*.) Liver trNK cells could be dissimilar from liver and splenic cNK cells perhaps due to their 'immature' phenotype, that is, CD27$^+$CD11b$^{low}$ (*Figure 2B*) and TRAIL$^+$ (*Figure 2C*). To further explore this possibility, we obtained transcriptome RNA-seq profiles from 'immature' bone marrow (BM) CD49a$^+$DX5$^-$ and CD49a$^-$DX5$^+$ NK cell samples and extracted CD49a$^+$DX5$^-$ and CD49a$^-$DX5$^+$-specific signatures. When we compared the levels of genes specific to immature CD49a$^+$DX5$^-$ NK cell in the BM (*Supplementary file 1*) to levels of these genes in the other cell subsets, we found that the CD49a$^+$DX5$^-$ cells in the liver did not display this cell-type specific pattern (*Figure 2D*). Since there was some variability in the relative expression levels of individual genes, further examination was performed by gene set enrichment analysis (GSEA) in which the genes differentially expressed in the DX5$^-$ BM subset was compared to the other NK cell subsets and plotted (*Figure 2—figure supplement 2*). Similar analysis was done for genes differentially expressed in the DX5$^+$ BM subset. These relative comparisons to the DX5$^-$ subsets showed that the CD49a$^-$DX5$^+$ subsets in BM, liver and spleen were similar to each other (p=0.01). However, the CD49a$^+$DX5$^-$ BM NK cells and liver CD49a$^+$DX5$^-$ trNK cells were dissimilar (p=0.5) (*Figure 2—figure supplement 2*). Taken together with our previous analysis with a much more limited panel of cell surface markers (*Peng et al., 2013*), these transcriptome data strongly suggest that the trNK cells in the liver are not a population of immature NK cells that are transitioning to mature cNK cells rather they represent a separate lineage of NK cells and indicate the CD49a$^-$DX5$^+$ subsets in BM, liver, and spleen likely represent the same recirculating population of cNK cells.

## Liver trNK cells are phenotypically distinct from cNK cells

To further analyze the similarities and differences between the NK cells of interest with greater precision, we examined their detailed phenotypes by flow cytometry. First, we examined the expressed repertoire of Ly49 receptors which are stably expressed on peripheral NK cells. The liver trNK cells express Ly49A, C/I, I, F, and G2 at much lower frequencies than cNK cells from both the liver and spleen (*Figure 3A,B*). Indeed, some receptors, such as the Ly49D and Ly49H activation receptors, are not expressed on trNK cells. However the frequencies of liver trNK cells expressing Ly49E (Ly49EF$^+$ but Ly49F$^-$) and NKG2A are much higher than liver and splenic cNK cells. In addition, the liver trNK cells appear larger and more granular by scatter parameters when compared to the liver and splenic cNK cells (*Figure 3C*). Since these parameters are associated with activated cNK cells, we examined expression of other activation markers. In naïve mice, the liver trNK cells express higher levels of CD69, CD44, CD160 and do not express CD62L as compared to the cNK cells in the liver and spleen, consistent with a more activated state.

To assess the cytokine profile of the liver trNK cells, we stimulated total liver lymphocytes with PMA and ionomycin and measured cytokine production (*Figure 4A–C*). The liver trNK cells specifically made easily detectable tumor necrosis factor-α (TNFα) and granulocyte macrophage colony-stimulating factor (GM-CSF) while liver and splenic cNK cells produced these cytokines at much lower levels, if at all (*Figure 4A*). By contrast, the liver trNK cells made similar amounts of interferon-γ (IFNγ) when compared to liver and splenic cNK cells. Interestingly when we analyzed IFNγ and TNFα simultaneously, we found a significant proportion of the liver trNK cells were IFNγ$^+$TNFα$^+$ double producers (*Figure 4B,C*), a population essentially absent among the cNK cells from the liver and spleen. Finally, we examined the responses of liver trNK cells to YAC-1 targets (*Figure 4D*). Although both trNK and cNK cells showed degranulation, as evidenced by CD107a expression, the degranulating liver trNK cells also produced TNFα that was rarely seen among the responding cNK cells. Thus, liver trNK cells display phenotypic differences from liver and spleen cNK cells whereas both cNK cell populations are similar, confirming and extending the transcriptome analysis.

## Liver trNK cells have differential transcription factor requirements than cNK cells

We analyzed liver trNK cells in IL-15Rα-deficient mice which have a deficiency in splenic cNK cells (*Lodolce et al., 1998*). The development of all NK1.1$^+$ CD3$^-$ cells was negatively impacted by the

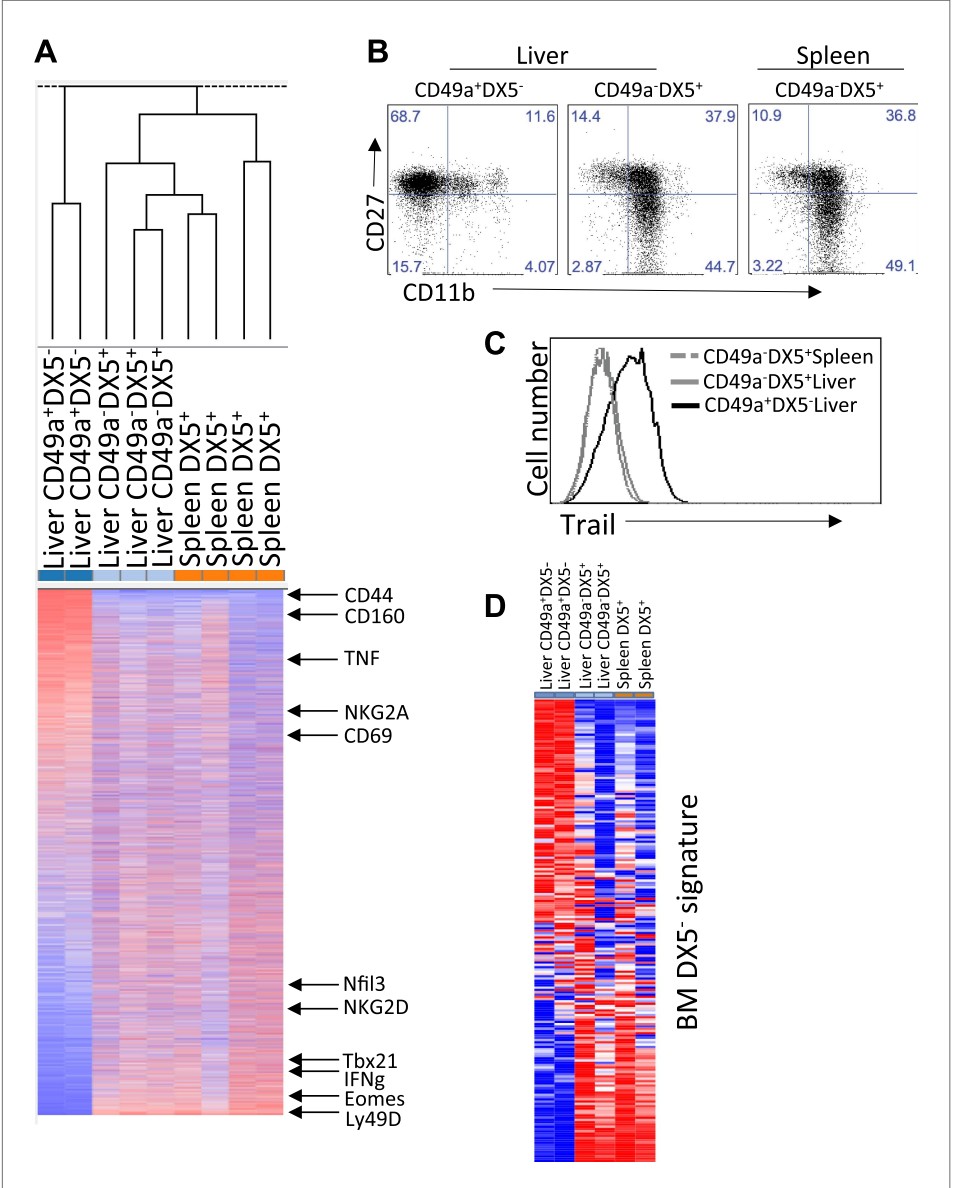

**Figure 2**. Distinct lineages of NK cells suggested by RNA-seq analysis. (**A**) Heat map showing cluster analysis of the entire gene set between the liver trNK cells and the cNK cells from the liver and spleen. We obtained expression profiles from sorted CD49a⁺DX5⁻ NK cells in the liver and CD49a⁻DX5⁺ cNK cells in the liver and spleen using the same small input RNA-seq approach. (For a heat map of a smaller number of genes highly expressed in liver trNK cells, see *Figure 2—figure supplement 1*.) (**B**) The trNK cells in liver display an 'immature' phenotype by flow cytometry. Cells from the liver and spleen were isolated, stained, and flow cytometry was performed. Dot plots were gated on live CD3⁻CD19⁻NK1.1⁺ cells and percentages in each dot plot represent the percentages of the subpopulations, CD49a⁺DX5⁻ cells in the liver and CD49a⁻DX5⁺ in the liver and spleen, that express CD11b and CD27. Dot plot profiles are representative of at least three experiments. (**C**) Liver trNK CD49a⁺DX5⁻ cells express TRAIL. Spleens and livers were isolated from WT C57BL/6NCr mice, stained, and flow cytometry was performed. The histogram was gated on live CD3⁻CD19⁻NK1.1⁺ cells and displays the expression level of TRAIL on CD49a⁺DX5⁻ liver trNK cells and CD49a⁻DX5⁺ cNK cells in the liver and spleen. (**D**) Expression of genes specific to DX5⁻ population of BM cells in DX5⁻ liver NK cells and DX5⁺ liver and spleen cNK cells shows non-specific pattern. Shown are ~200 genes most highly expressed in DX5⁻ BM NK cells as compared to DX5⁺ BM NK cells (genes shown in *Supplementary file 1*). Approximately half of the DX5⁻ BM NK cell-specific genes are upregulated in the liver CD49a⁺DX5⁻ NK cells and the other half is upregulated in the CD49a⁻DX5⁺ BM NK cells.
*Figure 2. Continued on next page*

*Figure 2. Continued*

Gene set enrichment analysis also shows non-significant relationship between BM DX5⁻ cells and CD49a⁺ NK cells (*Figure 2—figure supplement 2*).

The following figure supplements are available for figure 2:

**Figure supplement 1**. Heat map showing the top differentially expressed genes between the liver trNK cells and the cNK cells from the liver and spleen.

**Figure supplement 2**. Gene set enrichment analysis (GSEA) indicates relationships between BM NK cell and cNK cells in liver and spleen.

IL-15Rα deficiency, resulting in an absence of liver trNK cells as well as liver and splenic cNK cells (*Figure 5A*). Thus, despite their differences, the requirement for IL-15Rα suggests that the trNK cells and cNK cells are more related to each other than the non-NK cell members of the ILC family that do not require IL-15 (*Spits and Cupedo, 2012*; *Spits et al., 2013*).

While our RNA-seq and phenotypic analysis strongly suggest that the liver trNK cells represent a different NK cell lineage from cNK cells, an alternative hypothesis is that the liver trNK cells could be immature NK cells that ultimately become cNK cells, as suggested by their 'immature' phenotype (*Figure 2B*) and previous publications (*Townsend et al., 2004*; *Gordon et al., 2012*; *Peng et al., 2013*). To directly test these hypotheses and gain insight into the developmental relationship of liver

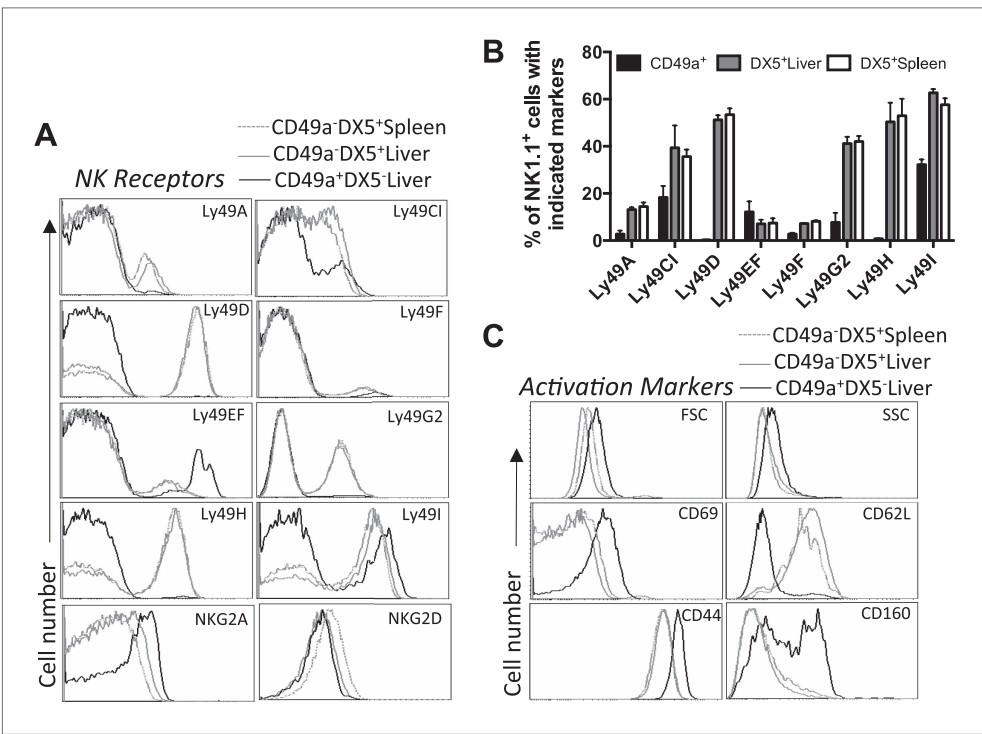

**Figure 3**. Liver trNK cells are phenotypically distinct from cNK cells. (**A** and **B**) Differential expression of NK cell receptors on liver trNK and splenic and liver cNK cells. Spleens and livers were isolated from WT C57BL/6NCr mice, stained, and flow cytometry performed. The histograms were gated on live CD3⁻CD19⁻NK1.1⁺ cells and display the expression level of NK receptors on CD49a⁺DX5⁻ liver trNK cells and CD49a⁻DX5⁺ cNK cells in the liver and spleen (**A**). A summary bar graph of the percentage of CD49a⁺DX5⁻ liver trNK cells and CD49a⁻DX5⁺ cNK cells in the liver and spleen that express the various NK receptors (**B**). (**C**) The liver trNK cells display an activated phenotype. Spleens and livers were isolated from WT C57BL/6NCr mice, stained, and flow cytometry performed. The histograms were gated on live CD3⁻CD19⁻NK1.1⁺ cells and display the expression level of activation markers that are expressed on CD49a⁺DX5⁻ liver trNK cells and CD49a⁻DX5⁺ cNK cells in the liver and spleen.

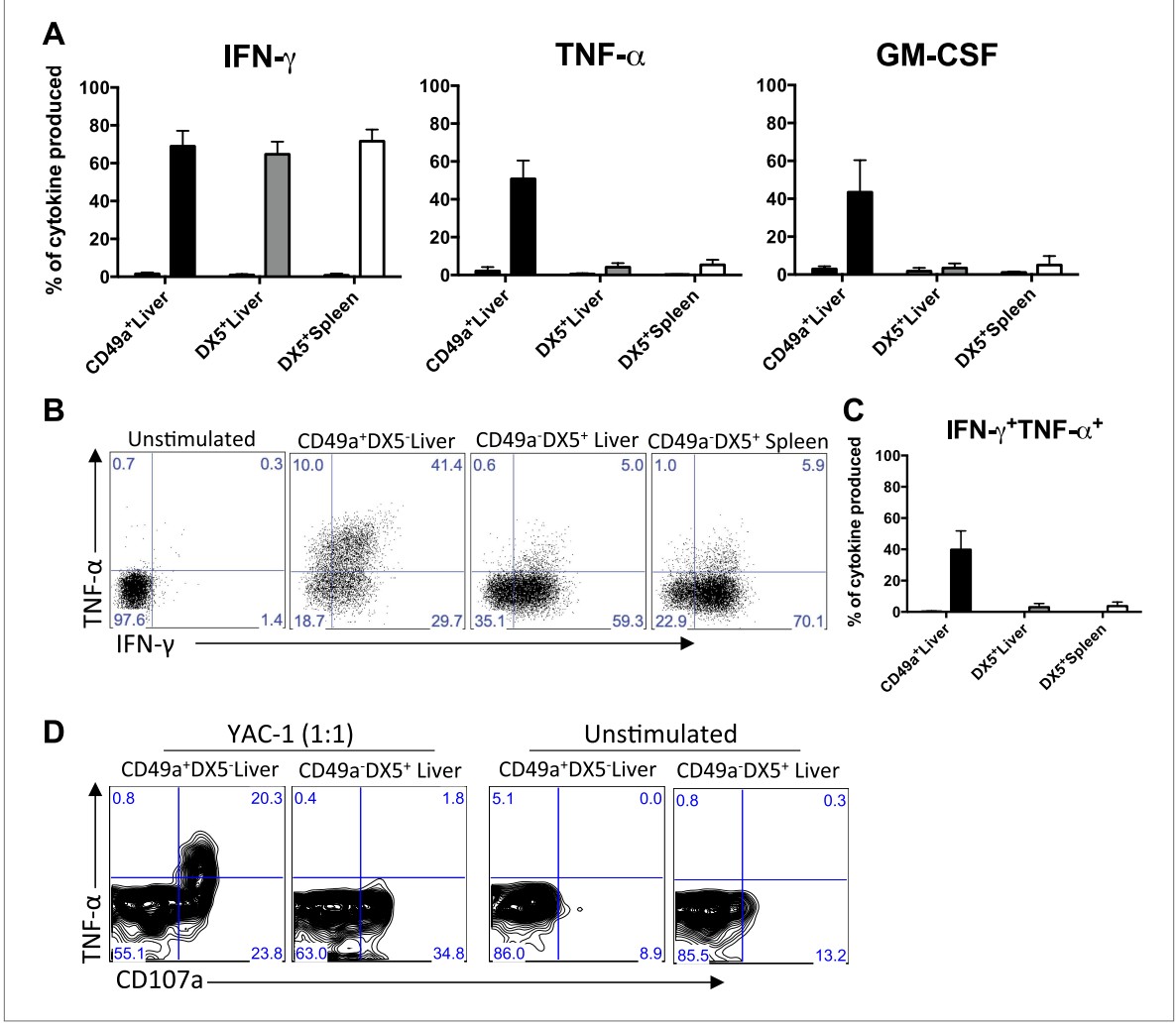

**Figure 4**. CD49a + DX5- trNK cells of the liver have a unique cytokine profile. (**A**) The liver trNK cells differentially express cytokines when activated. Spleens and livers were isolated from WT C57BL/6NCr mice stimulated with PMA/ionomycin for 4 hr, stained, and IFNγ, TNFα, and GM-CSF cytokine production was analyzed by flow cytometry. The cells were gated on live CD3⁻CD19⁻NK1.1⁺ and the graphs represent the percentage of indicated cytokine produced by CD49a⁺DX5⁻ liver trNK cells and CD49a⁻DX5⁺ cNK cells in the liver and spleen 4 hr post stimulation. (**B** and **C**) The liver trNK cells produce both IFNγ and TNFα when stimulated. Shown are dot plots of cells prepared and stimulated as in **A**. Each dot plot (**B**) was gated on live CD3⁻CD19⁻NK1.1⁺ cells and further gated on CD49a⁺DX5⁻ liver trNK cells and CD49a⁻DX5⁺ cNK cells in the liver and spleen. Cells were co-stained for IFNγ and TNFα and the percentage produced of each cytokine is presented in each quadrant. Bar graphs (**C**) indicate the percentage of IFNγ and TNFα-double producers shown in **B**. (**D**) The liver trNK cells degranulate and produce TNFα upon stimulation with YAC-1 targets. Liver lymphocytes were co-cultured at a 1:1 ratio with YAC-1 target cells for 6 hr. The cells were stained for the indicated markers and flow cytometry performed. The dot plots were gated on live CD3⁻CD19⁻NK1.1⁺ CD49a⁺DX5⁻ liver trNK cells or CD3⁻CD19⁻NK1.1⁺ CD49a⁻DX5⁺ cNK cells in the liver for CD107a degranulation and TNFα production.

trNK cells to cNK cells, we analyzed the mRNA levels of several transcription factors involved in cNK cell development. Particularly informative was decreased transcript levels of eomesodermin (Eomes) in the liver trNK cells as compared to liver and splenic cNK cells (*Figure 5B*). The differences in Eomes transcript levels were verified by flow cytometry which showed lower levels of Eomes protein in liver trNK as compared to cNK cells (*Figure 5C*), consistent with previous reports (*Gordon et al., 2012*). By contrast, there was no relative difference between liver trNK cells and liver and splenic cNK cells in Tbx21 (Tbet) expression at either the transcript or protein level (*Figure 5B,C*). Transcription factor staining was performed as previously described for Tbx21 (*Sojka and Fowell, 2011*). Thus, liver trNK cells express different levels of Eomes compared to spleen cNK cells, suggesting different

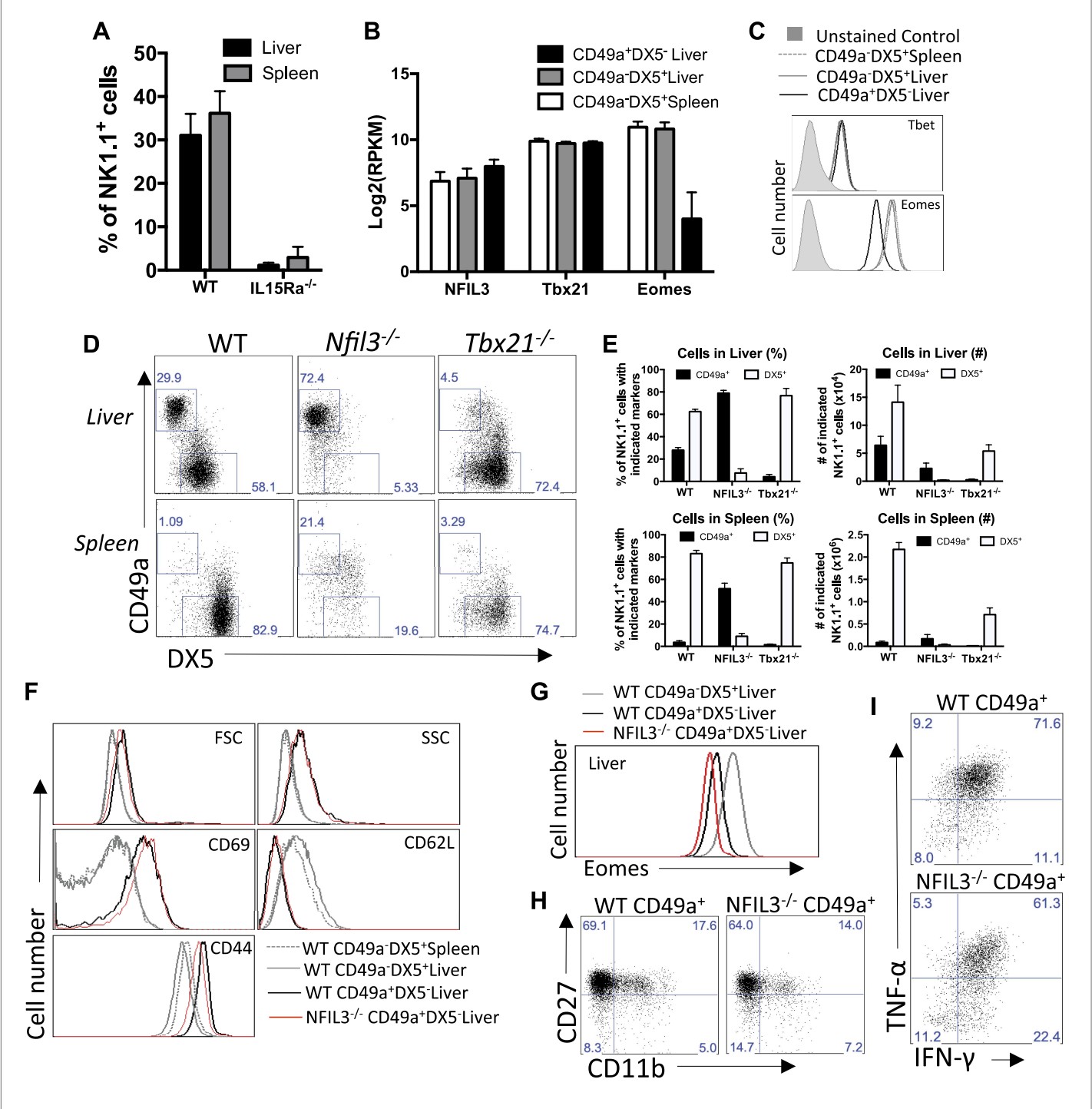

**Figure 5**. Liver trNK cells have different transcription factor requirements than cNK cells. (**A**) All liver NK cells require IL-15Rα. Spleens and livers were isolated from WT mice and *Il15ra*[−/−] mice, stained, and flow cytometry performed. The bar graph displays the percentage of CD3[−]CD19[−]NK1.1[+] cells in the liver and spleen of each strain of mice. (**B**) Eomes transcripts are expressed at lower levels in liver trNK cells. Spleens and livers were isolated from *Rag1*[−/−] mice and NK1.1[+] cells sorted for CD49a[+]DX5[−] liver trNK and liver and spleen CD49a[−]DX5[+] cNK cells. RNA-seq was performed on the sorted populations and the expression levels of indicated transcription factors plotted. RPKM = reads per kilobase per million mapped reads. Normalization of read counts by length of transcripts allowed comparison of expression levels of different genes. One of two independent experiments. (**C**) Eomes protein expression is decreased in liver trNK cells. Spleens and livers were isolated from *Rag1*[−/−] mice, stained, and flow cytometry performed for indicated transcription factors. The histograms were gated on live CD3[−]CD19[−]NK1.1[+] cells and display the expression level of transcription factors
*Figure 5. Continued on next page*

*Figure 5. Continued*

expressed in CD49a⁺DX5⁻ liver trNK cells and CD49a⁻DX5⁺ cNK cells in the liver and spleen. Histogram plots are representative of three independent experiments. (**D** and **E**) The liver trNK cells are present in NFIL3-deficient mice but absent in Tbx21 (Tbet)-deficient mice. Spleens and livers were isolated from WT, *Nfil3⁻/⁻*, and *Tbx21⁻/⁻* mice, stained, and flow cytometry performed. Representative dot plots (**D**) were gated on live CD3⁻CD19⁻NK1.1⁺ and display the expression level of CD49a and DX5 in the liver (top panels) and the spleen (bottom panels). Bar graphs (**E**) display the percentages (left column) and total NK cell number (right column) of CD3⁻CD19⁻NK1.1⁺ cells that express CD49a and DX5 in the liver and spleen of WT, *Nfil3⁻/⁻*, and *Tbx21⁻/⁻* mice. (**F**) The *Nfil3⁻/⁻* liver trNK cells display an activated phenotype, like liver trNK cells in WT mice. Spleens and livers were isolated from WT C57BL/6NCr and *Nfil3⁻/⁻* mice, stained, and flow cytometry was performed. The histograms were gated on live CD3⁻CD19⁻NK1.1⁺ cells and display the expression level of activation markers expressed on CD49a⁺DX5⁻ liver trNK cells and CD49a⁻DX5⁺ cNK cells in the liver and spleen in WT compared to the trNK cells from the *Nfil3⁻/⁻* mice. (**G**) *Nfil3⁻/⁻* liver trNK CD49a⁺DX5⁻ cells do not express Eomesodermin. Livers were isolated from WT C57BL/6NCr and *Nfil3⁻/⁻* mice, stained, and flow cytometry was performed. The histogram was gated on live CD3⁻CD19⁻NK1.1⁺ cells and displays the expression level of Eomes on WT and *Nfil3⁻/⁻* CD49a⁺DX5⁻ liver trNK cells and CD49a⁻DX5⁺ cNK cells in the liver. (**H**) The *Nfil3⁻/⁻* trNK cells in liver display an 'immature' phenotype by flow cytometry, similar to trNK cells in WT mice. Cells from the liver were isolated, stained, and flow cytometry was performed. Dot plots were gated on live CD3⁻CD19⁻NK1.1⁺ cells and numbers in each dot plot represent the percentages of the subpopulations, that is, liver CD49a⁺DX5⁻ cells and CD49a⁻DX5⁺ that express CD11b and CD27. Dot plot profiles are representative of two experiments. (**I**) Stimulated *Nfil3⁻/⁻* liver trNK cells produce cytokines similar to WT liver trNK cells. Livers were isolated from WT C57BL/6NCr and *Nfil3⁻/⁻* mice stimulated with PMA/ionomycin for 4 hr, and cells were co-stained for IFNγ and TNFα and analyzed by flow cytometry. The graphs were gated on live CD3⁻CD19⁻NK1.1⁺ cells and represent the percentage of cytokine⁺ (or cytokine⁻) cells among the CD49a⁺DX5⁻ liver trNK cells from WT and *Nfil3⁻/⁻* mice, as indicated. Dot plot profiles are representative of two experiments.

requirements for the related Tbox transcription factor, Tbet, which plays an overlapping role with Eomes in cNK cell development (*Gordon et al., 2012*).

To directly test specific transcription factor requirements, and to gain additional insight into the developmental relationship between the liver trNK cells vs liver and spleen cNK cells, we analyzed these cells in mice lacking NFIL3 and Tbet (Tbx21). Surprisingly, in NFIL3-deficient mice, liver trNK cells were present in total cell number (*Figure 5D,E*). Conversely, as previously reported, NFIL3-deficient mice had no cNK cells in the spleen (*Gascoyne et al., 2009*; *Kamizono et al., 2009*; *Kashiwada et al., 2010*). As well, there were no cNK cells in the liver (*Figure 5D,E*), consistent with their relationship to splenic cNK cells and raising the percentage of liver trNK cells. By contrast, liver trNK cells were absent in Tbet-deficient mice in both percentage and total cell number, when compared to WT controls and in contradistinction to both cNK cell populations (*Figure 5D,E*). The requirement for Tbet is consistent with an intrinsic requirement for Tbet in development of TRAIL⁺ liver NK cells (*Gordon et al., 2012*). Thus, these data demonstrate a unique developmental pathway of liver trNK cells that does not require NFIL3 but depends on Tbx21, indicating that the liver trNK cells are not precursors of cNK cells.

To assess the phenotype and function of the trNK cells in the liver of the NFIL3-deficient mice we assessed the cells at steady state and upon activation by flow cytometry. We found that much like the liver trNK cells from a WT mouse, the trNK cells from *Nfil3⁻/⁻* mice also express markers correlated with activation, that is, higher FSC and SSC, higher levels of CD69 and CD44 and lower levels of CD62L (*Figure 5F*) and do not express Eomes (*Figure 5G*). They have an 'immature' phenotype similar to the WT trNK cells in that they express CD27 and generally lack CD11b expression on most cells (*Figure 5H*). Upon stimulation, the *Nfil3⁻/⁻* trNK cells in the liver produce both IFNγ⁺ and TNFα⁺ simultaneously in a manner comparable to WT trNK cells (*Figure 5I*). Thus, NFIL3 is not required for normal trNK cells in the liver.

Taken together, these data indicate that liver trNK cells require different transcription factors than liver and splenic cNK cells, providing strong evidence that the liver trNK cells are a distinct lineage from the liver and splenic cNK cells, whereas both cNK cell populations are likely to be developmentally identical.

## Tissue-resident NK cells in other organs

Inasmuch as NK cells have been described in other solid organs, we next determined if CD49a could also mark trNK cells in other tissues. In addition to the liver, we found large percentages of CD49a⁺DX5⁻ NK cells in the uterus and skin but not in other organs examined (*Figure 6A,B*), suggesting that CD49a may be a marker of trNK cells in these organs. The CD49a⁺ NK cells in the uterus and skin also generally lacked expression of DX5 (*Figure 6B*) and displayed constitutive expression of CD69, much like the liver trNK cells (*Figure 6C*). To determine if the CD49a⁺DX5⁻ cells were resident to the uterus and skin, we studied parabiotic mice as we had done previously to help establish the tissue residency of CD49a⁺ DX5⁻ NK cells in the liver (*Peng et al., 2013*). At 2 weeks post surgery, we assessed both the trNK and cNK cells in the uterus and skin in each

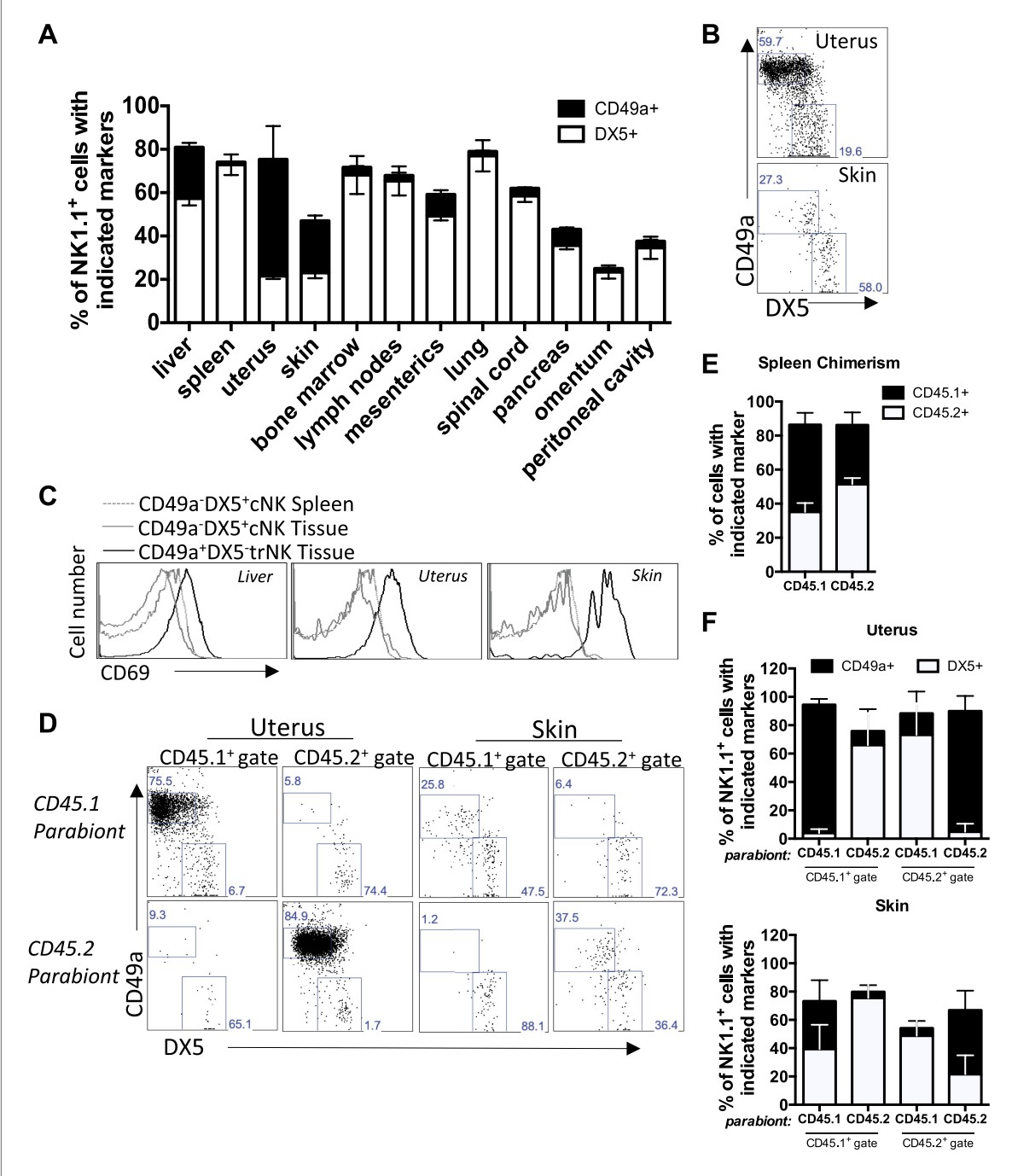

**Figure 6**. Tissue-resident NK cells in other organs. (**A**) CD49a+DX5− NK cells are present in liver, skin, and uterus. Various organs were isolated from WT C57BL/6NCr mice, stained, and flow cytometry performed. The stacked bar graph represents the percentage of live CD3−CD19−NK1.1+ cells that either express CD49a or DX5 in the indicated organs. (**B**) CD49a expression on skin and uterine trNK cells. The dot plot displays cells that were isolated from the uterus and skin of WT C57BL/6NCr mice. The dot plots were gated on live CD3−CD19−NK1.1+ cells and the percentage of cells expressing CD49a and DX5 are shown. (**C**) CD49a+DX5− NK cells in liver, skin, and uterus express higher levels of CD69. Spleen, liver, uterus, and skin were isolated from WT C57BL/6NCr mice, stained, and flow cytometry performed. The histograms were gated on live CD3−CD19−NK1.1+ cells and display the expression level of CD69 on CD49a+DX5− trNK cells and CD49a−DX5+ cNK cells in the spleen, and indicated tissues. (**D** and **F**) CD49a+DX5− NK cells in uterus and skin are tissue-resident as revealed by parabiotic mice. The uterus (left two panels) and skin (right two panels) were isolated from day 14 parabiosed mice. (**E**) Chimerism in the spleen. The spleen was analyzed for the degree of chimerism by analyzing the percentage of gated live CD45.1 and CD45.2 cells in each parabiont on day 14-post parabiosis surgery. These data correspond to data shown in *Figure 6D,F*. (**F**) Each parabiont was analyzed for its host and migratory cells using the congenic markers CD45.1 and CD45.2. The cells were

*Figure 6. Continued on next page*

*Figure 6. Continued*

further gated on live CD3⁻CD19⁻NK1.1⁺ cells and the percentages of cells expressing CD49a and DX5 are shown in the representative dot plots (**D**). Stacked bar graphs (**F**) show the percentages of cells expressing CD49a and DX5, the gated populations in (**D**). The experiment was performed two independent times with four parabiosed animals in each experiment (total of 8 pairs).

parabiont by flow cytometry by gating on the appropriate CD45.1 or CD45.2 congenic marker in order to differentiate host-derived cells from circulating cells from the other parabiont. In both the uterus and skin, host CD49a⁺ DX5⁻ NK cells were primarily found in the indicated host tissue (*Figure 6D,F*), similar to the CD49a⁺ DX5⁻ trNK cells in the liver (*Peng et al., 2013*). By contrast, CD45 allotype-disparate mice reached nearly complete chimerism in the spleen (*Figure 6E*) (*Wright et al., 2001*; *Peng et al., 2013*). Moreover, the CD49a⁻DX5⁺ NK cells of both allotypes were found in both parabionts, indicating that they were circulating NK cells, akin to the CD49a⁻DX5⁺ cNK cells in the liver (*Figure 6D,F*). Thus, the uterus and skin contain both trNK (CD49a⁺ DX5⁻) and circulating cNK (CD49a⁻DX5⁺) cNK cells, with the trNK cells appearing to dominate the uterine NK cell population.

Although uterine NK cells have not been studied previously in the context of trNK cells, they are known to be IL-15-dependent (*Ashkar et al., 2003*) and thus IL-15Rα-dependent. Since little is known about NK cells in the skin, we analyzed them further. The skin trNK cells do not express Eomes and are absent in IL-15Rα-deficient mice (*Figure 7A,B*). Furthermore the trNK cells in the uterus and skin produce more IL-2 than cNK cells in the liver and spleen (*Figure 7C*). These features are similar to liver trNK cells.

We evaluated whether the trNK cells in the uterus and skin are related to the CD127⁺ thymic NK by two independent ways. First, we found no detectable CD127 expression on uterine and skin trNK cells when compared to the CD127 expression on the thymic NK cells (*Figure 7D*). Second, we found uterine and skin trNK cells are present in Foxn1⁻/⁻(athymic) mice (*Figure 7E*), indicating that they develop independent of a thymus, unlike thymic NK cells.

To determine if the trNK cells in the liver, uterus, and skin are related lineages, we examined the NFIL3- and Tbet-deficient mice. Strikingly, trNK cells were present in the liver, uterus and skin of NFIL3-deficient mice (*Figure 8A,B*), indicating that these trNK cells represent a distinct lineage from cNK cells. Moreover, the circulating CD49a⁻DX5⁺ NK cells were absent in these tissues, indicating that these CD49a⁻DX5⁺ NK cells are related to the cNK cells in the liver and spleen. In Tbet-deficient mice, the trNK cells were absent in skin (*Figure 8A,B*) akin to liver trNK cells (*Figure 5D,E*), implying a developmental link between the trNK cells in the liver and skin. Interestingly, however, the uterine trNK cells were still present in Tbet-deficient mice (*Figure 8A,B*), strongly suggesting that uterine trNK cells represent yet another NK cell lineage.

## Discussion

Many immune cell lineages migrate throughout the body via the circulatory system. However, emerging data indicate that a number of different immune cell types appear to be tissue-resident and rarely recirculate. Here we demonstrate that NK cells are also comprised of circulating and several tissue-resident cell types in the liver, skin and uterus. Our comprehensive transcriptome and FACS analyses on liver trNK cells and liver and splenic cNK cells suggested that they may be distinct lineages of NK cells. Indeed, circulating liver and splenic cNK cells were absent in NFIL3-deficient mice, further demonstrating their lineage relationship to each other. On the other hand, NFIL3-deficient mice still possessed trNK cells in the liver, skin, and uterus. Moreover, Tbet-deficient mice lacked trNK cells in the liver and skin but uterine trNK cells and cNK cells in the liver and spleen were largely intact. Finally, thymic NK cells represent a distinct NK cell lineage because they are absent in nude and GATA-3-deficient mice which generally possess cNK and trNK cells. Taken together, these data suggest that there are at least four lineages of NK cells: cNK cells circulating in spleen, blood and other organs, thymic, and two trNK cell lineages—liver (and skin), and uterine.

The trNK cells can be distinguished from cNK cells in several different ways. Most importantly, trNK cells do not require the putative NK cell specification factor, NFIL3 (*Di Santo, 2009*; *Gascoyne et al., 2009*; *Kamizono et al., 2009*; *Kashiwada et al., 2010*), indicating that they belong to a different cell

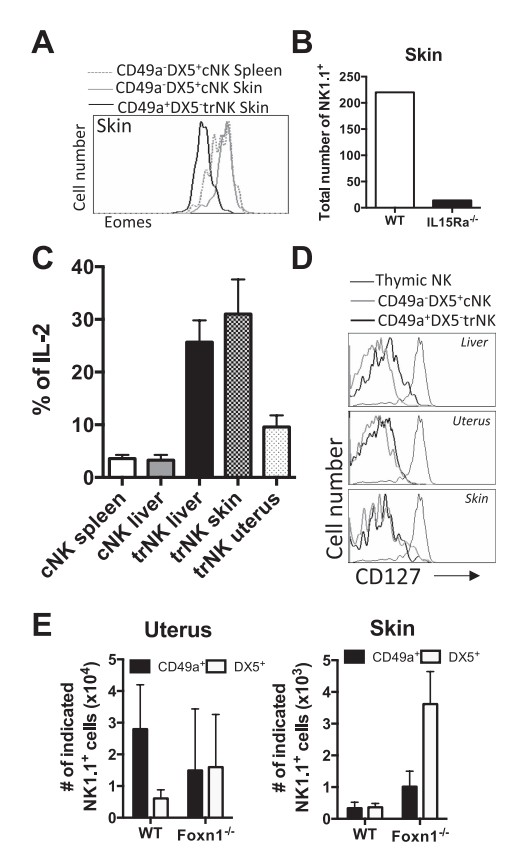

**Figure 7**. trNK cells in the liver resemble trNK cells in the skin and not the uterus, and all develop independent of a thymus. (**A**) Eomes is poorly expressed in skin trNK cells. Skin was isolated from WT C57BL/6NCr mice, stained, and flow cytometry performed. The histogram was gated on live CD3⁻CD19⁻NK1.1⁺ cells and displays the expression level of Eomes on CD49a⁺DX5⁻ skin trNK cells and CD49a⁻DX5⁺ cNK cells in the skin and spleen. (**B**) Skin trNK cells are absent in IL-15Rα-deficient mice. Skin was isolated from WT mice and *Il15ra⁻/⁻* mice, stained, and flow cytometry performed. The bar graph displays the number of events captured in the CD3⁻CD19⁻NK1.1⁺ gate. (**C**) CD49a⁺DX5⁻ trNK cells of the liver and skin produce IL-2. Spleen, liver, uterus and skin were isolated from WT C57BL/6NCr mice stimulated with PMA/ionomycin for 4 hr, stained, and IL-2 production was analyzed by intracellular staining and flow cytometry. To obtain the percentage of IL-2⁺ cells in each population, the graphs were derived from gated live CD3⁻CD19⁻NK1.1⁺ cells and represent the percentage of IL2⁺ cells among the CD49a⁻DX5⁺ cNK cells in the liver and spleen and the CD49a⁺DX5⁻ liver, uterus and skin trNK cells. (**D**) Tissue-resident CD49a⁺DX5⁻ cells in the liver, uterus and skin do not express CD127. Thymus, liver, uterus and skin were isolated from WT C57BL/6NCr mice, stained, and flow cytometry was performed. The histograms are from gated live CD3⁻CD19⁻NK1.1⁺ cells and display the expression levels of CD127 on CD49a⁺DX5⁻ liver, uterus

*Figure 7. Continued on next page*

lineage. In the liver, skin, and uterus, they differentially express CD49a, though it is possible that trNK cells in other organs preferentially express another marker. Concomitantly, they lack DX5 expression which is generally expressed as a late maturation marker on most cNK cells (*Kim et al., 2002*; *Peng et al., 2013*). Interestingly, trNK cells from unmanipulated mice display markers associated with cNK cell activation, such as CD69, in contrast to cNK cells which do not express CD69 until activated (*Karlhofer and Yokoyama, 1991*; *Wang et al., 2000*). The trNK cells also express a different repertoire of Ly49 receptors, suggesting that they may be tolerant to self by using mechanisms that do not strictly employ MHC class I-dependent licensing by Ly49 receptors, as has been shown for cNK cells in the spleen (*Kim et al., 2005*; *Elliott and Yokoyama, 2011*). The trNK cells efficiently produce other cytokines, particularly TNFα and GM-CSF which may contribute to inflammatory conditions in a manner distinct from cNK cells which predominantly produce IFNγ. TNFα production occurs following stimulation by target cells and also occurs in trNK cells producing IFNγ. Finally, trNK cells utilize a separate set of transcription factors which may endow trNK cells with other unique functions, in addition to lineage commitment. As such, due to their tissue localization and these distinctive features, trNK cells most likely play important roles in innate defense in ways that should be distinguishable from cNK cells. Future investigations to re-examine the contribution of NK cells in different organs with a focus on trNK cells should reveal these roles.

Among the various NK cell lineages, the liver and skin trNK cells appear to be highly related. They are phenotypically similar and both are absent in Tbet-deficient mice. Although it remains to be formally shown that a common precursor can give rise to liver and skin trNK cells but not other NK cell lineages, these data indicate that they are more related to each other than to other NK cell lineages, including cNK cells. uNK cells appear distinct from the liver and skin trNK cells because they are Tbet-independent, even though all are NFIL3-independent. Perhaps this is due to concomitant expression of Eomes at high levels in uNK cells (*Tayade et al., 2005*). Alternatively, uterine trNK cells may depend on other transcription factors unique for this NK cell lineage.

The liver, skin, and uterine trNK cell lineages can be distinguished from thymic NK cells in several ways. Both trNK (as shown here) and cNK cells do not express high levels of CD127 that is characteristic of thymic NK cells (*Vosshenrich et al., 2006*; *Yadi et al., 2008*). Moreover, thymic

*Figure 7. Continued*

and skin trNK cells and CD49a⁻DX5⁺ cNK cells in the liver, uterus and skin and NK1.1⁺ thymic NK cells. (**E**) Tissue-resident CD49a⁺DX5⁻ cells in the liver, uterus and skin develop independent of a thymus .The uterus and skin were isolated from WT and *Foxn1⁻/⁻* mice, stained, and flow cytometry was performed. Bar graphs display the total number of CD3⁻CD19⁻NK1.1⁺ cells that express CD49a and DX5 in the uterus and skin of WT and *Foxn1⁻/⁻* mice. Experiments were performed three independent times.

NK cells are absent in nude mice and in mice lacking GATA-3 which possess cNK cells, as recapitulated here. By contrast, we showed liver trNK cells are present in nude and GATA-3 deficient mice, suggesting that skin trNK cells will show these features, although the skin and uterine trNK cells need to be further studied in these mice. The trNK cells in the uterus in virgin mice as studied here will also need to be studied with respect to other subsets of uNK cells which have been described previously, including endometrial vs decidual NK cells, and NK1.1⁻ DX5⁻ uterine NK cells (*Yadi et al., 2008*; *Mallidi et al., 2009*).

Inasmuch as liver, skin, and uterine trNK cells can be distinguished from cNK cells, it is important to note that they do share some important similarities, in addition to surface expression of NK1.1 and NKp46 but not CD3ε. As shown here and previously published, cNK, uterine and liver trNK cells can respond to YAC-1 targets by degranulation and killing (*Kiessling et al., 1975*; *Linnemeyer and Pollack, 1991*; *Peng et al., 2013*), though this needs to be further detailed for the skin trNK cells described here. Moreover, cNK cells require IL-15 or IL-15Rα (*Lodolce et al., 1998*; *Kennedy et al., 2000*) and here we showed that IL-15Rα-deficient mice have no cNK or trNK cells, such that no NK1.1⁺ CD3⁻ cells were identifiable in spleen, liver, and skin of the knockout mice. Similarly, IL-15-deficient mice lack NK1.1⁺ CD3⁻ cells in the uterus (*Ashkar et al., 2003*; *Barber and Pollard, 2003*), indicating that trNK cells in the uterus as well as liver and skin are IL-15Rα-dependent. While more detailed analysis of the progenitors for NK cells will be required to establish the precise lineage relationship of trNK and cNK cells to each other, current data, particularly cytotoxic potential and IL-15-dependence, support their intimate relationship, despite differences in transcription factor requirements.

Recent studies indicate that cNK cells are related to innate lymphoid cells (ILCs) (*Spits and Cupedo, 2012*; *Spits et al., 2013*). While this area of research is fluid with potentially more cells to be identified and additional characteristics to be discovered, it now appears that ILCs can be separated into three groups, based on shared characteristics (*Spits et al., 2013*). cNK cells are now classified as belonging to the group 1 ILCs, due to their shared production of IFNγ. cNK cells can be distinguished from other group 1 ILCs by their cytotoxic capacities, dependence on IL-15, and general lack of IL7Rα expression, features shared between cNK and liver, skin, and uterine trNK cells. Recently, a new IFNγ-producing ILC1 cell was identified which demonstrated dependence on NFIL3, Tbx21, and IL-15 but not IL-15Rα (*Fuchs et al., 2013*), unlike the trNK cells described here. Finally, while the relationship of thymic NK cells expressing CD127 to ILCs which also express CD127 may need further refinement, current knowledge suggest that the liver (skin) and uterine trNK cells are more related to cNK cells than ILCs.

Future studies, however, may continue to blur the lines between ILCs and NK cells. Already, it has been noted that markers previously thought to be exclusively expressed on NK cells, such as NKp46, are expressed on ILCs that now can be distinguished from classical NK cells (*Walzer et al., 2007*; *Cella et al., 2009*; *Spits et al., 2013*). A further confounding issue is the apparent plasticity of ILCs with potential interconversion between ILC groups (*Cella et al., 2010*; *Vonarbourg et al., 2010*), indicating the increasing complexities of definitively identifying ILC types and potentially distinguishing ILCs from other immune cells. Thus, it is possible that trNK cells may be more closely related to certain ILCs than currently appreciated.

Tissue-resident NK cells are more closely related to ILCs in that both the trNK cells and the ILCs have extended their place of residency outside the traditional secondary lymphoid organs. Originally most ILCs were considered in the context of gut or lymphoid tissues (*Spits and Cupedo, 2012*) but recently they have been identified in other nonlymphoid tissues such as the skin and the uterus. ILC2s in the skin have been reported to regulate cutaneous inflammation but they are unaffected by IL15-deficiency (*Roediger et al., 2013*), unlike trNK cells as shown here. In the uterus, immature NK cells have been described as well as a phenotypically related but distinct population of cells that express the transcription factor RORC and produce IL-22, suggesting that they may be ILC3s rather than NK cells (*Crellin et al., 2010*; *Male et al., 2010*). Additional studies will be needed to more finely localize the anatomic location of trNK cells in these tissues and the transcription factors required

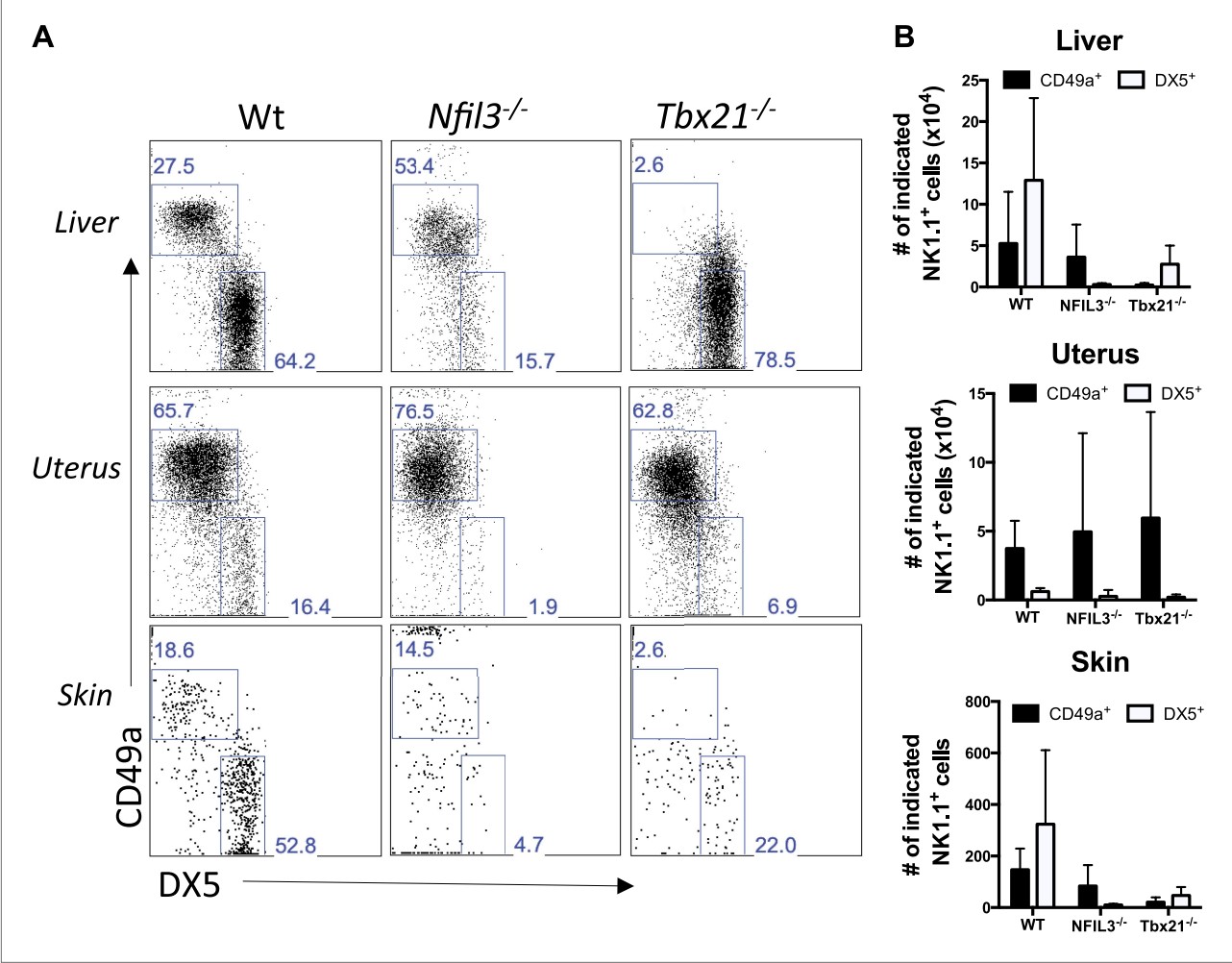

**Figure 8**. Liver, uterus, and skin trNK cells have different requirements for NFIL3 and Tbet (Tbx21). (**A**) The liver, uterus and skin were isolated from WT, *Nfil3*[-/-], and *Tbx21*[-/-] mice, stained, and flow cytometry performed. Representative dot plots were gated on live CD3−CD19−NK1.1+ cells and display the expression level of CD49a and DX5 in the liver (top panels) and the uterus (middle panels) and skin (bottom panels). Percentages indicate the gated populations. Bar graphs (**B**) display the total number of CD3−CD19−NK1.1+ cells that express CD49a and DX5 in the liver, uterus and skin of WT, *Nfil3*[-/-], and *Tbx21*[-/-] mice. Experiments were performed five independent times.

for their development and maintenance, work that will be aided by development of new tools to distinguish trNK cells from cNK cells and ILCs.

The trNK cells should also be considered in the context of other non-lymphoid tissue-resident immune cells, including NKT cells, tissue-resident memory T cells, γδ T cells, B1 B cells, and tissue macrophages, among others. Of these, NKT cells are of special interest because they share several characteristics with trNK cells in the liver. NKT cells were first identified because they express markers associated with NK cells, such as NK1.1, but are not cNK cells because they express rearranged TCR genes (*Bendelac et al., 2007*). While they are now known to recognize glycolipids presented by CD1 molecules, a large number of NKT cells also reside in the liver, though they can be found in other tissues. In the liver, NKT cells crawl within the sinusoidal space by patrolling sinusoidal endothelial cells (*Geissmann et al., 2005*). This is reminiscent of early electron microscopy studies of the rat liver which identified 'pit cells', now recognized as NK cells (*Wisse et al., 1976*; *Bouwens et al., 1987*). Pit cells are found in the sinusoidal space usually adjacent to sinusoidal endothelial cells. Although liver trNK cells have not been formally examined vis-à-vis pit cells, it seems likely that they are related, if not equivalent.

The trNK cell populations also are potentially related to tissue-resident memory T cells which had been previously activated by antigen-stimulation and differentiated into effector cells that then reside

in non-lymphoid tissues (*Sheridan and Lefrancois, 2011*; *Gebhardt et al., 2013*). Our studies indicate that trNK cells appear to have an activated phenotype with expression of CD69 and other activation markers as well as other phenotypic changes. However, it is not yet clear if trNK cells differentiate from a circulating precursor that then differentiates into a cell taking up tissue residency, akin to tissue-resident memory T cells (*Masopust et al., 2010*). Our studies do not rule out this possibility although the transcription factor requirements make it unlikely that a putative circulating precursor is an NK cell that had previously differentiated into a cNK cell.

An alternative possible origin of trNK cells is a progenitor that seeds the peripheral organs during embryonic or fetal life. In particular, due to early hematopoiesis occurring in the fetal liver, liver trNK cell precursors may seed the liver during embryonic life. In this regard, it is interesting to note that Ly49E was first noted to be almost exclusively expressed on fetal NK cells (*Toomey et al., 1998*). However, recent findings (*Filtjens et al., 2013*) and those reported here indicate that liver trNK cells in adult mice also selectively express Ly49E, raising the possibility that Ly49E⁺ NK cells in the fetus reflect trNK cells already present in the embryonic liver, at a time when cNK cells are poorly developed (*Bukowski et al., 1985*; *Dorfman and Raulet, 1998*). As such, since the liver is the major site for fetal hematopoiesis, Ly49E⁺ NK cells may not reflect fetal NK cells per se but instead the possibility that precursor of trNK cells seeds the liver during early life. Consistent with this possibility, we recently showed that precursors of liver trNK cells are present in the adult liver (*Peng et al., 2013*). Moreover, adult BM hematopoietic stem cells do not fully reconstitute the liver trNK cell population in irradiated mice. Thus, precursors of trNK cells, like γδT cells and certain tissue macrophages (*Havran and Allison, 1990*; *Schulz et al., 2012*), may seed tissues early in life, a topic that will require further evaluation.

In conclusion, we provide molecular evidence that trNK cells are distinct from circulating cNK cells in their cytokine profiles, expression of NK cell receptors and most importantly their transcription factor requirements. Their tissue residency feature suggests that they have tissue-specific homeostatic functions. Our studies should prompt a re-examination of the myriad roles of NK cells in many immune responses which could be due to trNK cells, rather than cNK cells. Finally, our studies may also be applicable to emerging data on other hematopoietic cells that circulate throughout the body in that there may be related but distinct cell types that are tissue-resident.

## Material and methods

### Mice

All mice were housed in a pathogen-free facility and procedures performed in accordance with the animal protocol approved by the Washington University School of Medicine (WUSM) Animal Studies Committee or the NIAID Animal Care and Use Committee. WT C57BL/6NCr and B6-LY5.2/Cr mice were purchased from the National Cancer Institute (Frederick, MD). *Foxn1*⁻/⁻ (nude), *Tbx21*⁻/⁻ (Tbet-deficient), *Il15ra*⁻/⁻ and *Rag1*⁻/⁻ were purchased from Jackson Laboratory (Bar Harbor, ME). The *Il15ra*⁻/⁻ were backcrossed to C57BL/6 background by speed congenics (*Wakeland et al., 1997*). *Nfil3*⁻/⁻ and *Gata3*ᶠˡ/ᶠˡ mice were previously described (*Zhu et al., 2004*; *Kashiwada et al., 2010*). *Gata3*ᶠˡ/ᶠˡ mice were bred onto C57BL/6 background for at least nine generations before they were bred with Vav-Cre transgenic mice (JAX line 8610) on a C57BL/6 background to generate *Gata3*ᶠˡ/ᶠˡ-Vav-Cre line.

### Cell isolation

Single cell suspensions were made from spleen, bone marrow, peripheral and mesenteric lymph nodes, spinal cord, omentum, peritoneal cavity, pancreas, lung, liver, skin, and uterus for flow cytometry analysis. Briefly, all organs isolated were mechanically dissociated and put through a 100 µm or 70 µm cell strainer. The liver single cell suspension was resuspended in 40% Percoll and centrifuged for 20 min at 2000 rpm at room temperature. Bone marrow was isolated from femurs. Spinal cord, omentum, pancreas, lung, uterus and skin were all incubated in 167 µg/ml liberase TL (Roche Applied Science, Indianapolis, IN) and 100 µg/ml of DNase (Roche Applied Science) for 1 hr at 37°C in shaking incubator. Skin lymphocytes were isolated from ears; each ear was split into dorsal and ventral sides prior to liberase treatment.

### Cell stimulation

Single cell suspensions of spleen and liver were cultured in 10% heat inactivated fetal calf serum and RPMI 1640 and stimulated with phorbol 12-myristate 13-acetate (PMA, 200 ng/ml) and Ionomycin (5 µg/ml) for 4 hr in the presence of Brefeldin A (BD Pharmingen, San Jose, CA) to analyze cytokine

production. For the degranulation assay, a suspension of $10^5$ liver cells was plated with YAC-1 target cells at an effector:target (E:T) ratio of 1:1 in 96-well V-bottom plates. Anti-CD107a antibody, Brefeldin A, and monensin (eBioscience, San Diego, CA) were added to each well before incubation. Plates were incubated for 6 hr at 37°C, after which surface staining followed by intracellular staining for TNFα was performed for analysis by flow cytometry.

## Antibodies and flow cytometry

All cells were stained with Fixable Viability Dye, prior to blocking Fc receptors with 2.4G2, per manufacturer's instructions (eBioscience). The cells were stained with cell surface antibodies. To measure cytokine production the cells were fixed and permeabilized using the BD cytofix/cytoperm reagents and the forkhead box P3 (Foxp3) fix/perm reagents were used to assay transcription factors (eBioscience). Events were acquired on the Canto (BD Biosciences) using the FACSDiva software (BD Biosciences) and analyzed with FlowJo software (Treestar).

The following antibodies were purchased from eBioscience: anti-CD3 (clone 145-2C11), anti-CD19 (eBio1D3), anti-NK1.1 (PK136), anti-CD49b (DX5), anti-CD27 (LG.7F9), anti-CD11b (M1/70), anti-CD69 (H1.2F3), anti-CD62L (MEL-14), anti-CD44 (IM7), anti-CD160 (eBioCNX46-3), anti-IFNγ (XMG1.2), anti-TNFα (MP6-XT22), anti-GM-CSF (MP1-22E9), anti-CD107a (eBio1D4B) anti-Eomesodermin (Dan11mag), anti-CD45.1 (A20), anti-CD45.2 (104), anti-TRAIL (N2B2), anti-CD127 (A7R34) anti-Ly49A (A1), anti-Ly49 E/F (CM4), anti-Ly49D (eBio4E5), anti-Ly49G2 (eBio4D11), anti-Ly49H (3D10), anti-Ly49I (YLI-90), anti-NKG2A (16a11), and anti-NKG2D (CX5). Antibodies purchased from BD were anti-CD49a (Ha31/8), anti-Ly49F (HBF-719) and anti-Ly49C/I (5E6). Anti-Tbet (4B10) was purchased from Biolegend (San Diego, CA) and used as previously described (*Sojka and Fowell, 2011*).

## Deep sequencing

Spleen, liver and bone marrow cells were isolated from $Rag1^{-/-}$ mice and sorted for CD3⁻CD19⁻NK1.1⁺ CD49a⁺DX5⁻ or CD3⁻CD19⁻NK1.1⁺CD49a⁻DX5⁺ which were lysed. mRNA was extracted from cell lysates using oligo-dT beads (Invitrogen, Grand Island, NY). For cDNA synthesis, we used custom oligo-dT primer with a barcode and adapter-linker sequence (CCTACACGACGCTCTTCCGATCT—XXXXXXXX-T15). After first strand synthesis samples were pooled together based on *Actb* qPCR values and RNA-DNA hybrids were degraded using consecutive acid-alkali treatment. Then, second sequencing linker (AGATCGGAAGAGCACACGTCTG) was ligated using T4 ligase (New England Biolabs, Ipswich, MA) and after SPRI clean-up, mixture was PCR enriched 14 cycles and SPRI purified to yield final strand specific 3'end RNA-seq libraries. Data were sequenced on HiSeq 2500 instrument (Illumina, San Diego, CA) using 50 bp × 25 bp pair-end sequencing. Second mate was used for sample demultiplexing, at which point individual single-end fastqs were aligned to mm9 genome using TopHat with following options -G mm9.mrna.10.31.gtf–prefilter-multihits–segment-length 20 –max-multihits 15. Gene expression was obtained using ESAT software tool (http://garberlab.umassmed.edu/software/esat/) focused on analysis of 3'end targeted RNA-Seq. The following parameters were used: task 'score3P', normalized-Output, window 1000, maxExtension 3000, maxIntoGene 2000, stranded, collapseIsoforms.

## Parabiosis

Parabiosis surgery was performed as previously described (*Peng et al., 2013*). Briefly, matching longitudinal skin incisions were made on the flanks of C57BL/6NCr (Ly5.2) and B6-LY5.1/Cr female mice. Their elbows and knees were then joined with dissolvable sutures and the incisions were closed with wound clips. Postoperative care included administration of buprenex compound for pain management, 5% dextrose and 0.9% sodium chloride. Nutritional gel packs were provided in each cage and antibiotics (Sulfatrim) in the drinking water for the duration of the experiment.

## Acknowledgements

We thank the Yokoyama lab for great discussions, Marco Colonna for critically reading the manuscript, and Dorjan Brinja and Erica Lantelme for cell sorting. We thank Michel Nussenzweig and his lab for initial help with parabiotic mice. This work was supported by NIH grants R01AI106561 and R01AI033903 and National Basic Research Project of China (973 project) (2013CB944902). CZ and JZ are supported by the Division of Intramural Research of the NIAID (US National Institutes of Health). The Rheumatic Diseases Core Center (P30AR048335) performed the speed congenics backcross. WMY is an Investigator of the Howard Hughes Medical Institute. DKS was supported by T32 CA009547.

## Additional information

### Funding

| Funder | Grant reference number | Author |
|---|---|---|
| Howard Hughes Medical Institute | | Wayne M Yokoyama |
| National Institutes of Health | 1R01AI106561 | Wayne M Yokoyama |
| National Institutes of Health | 2R01AI033903 | Wayne M Yokoyama |
| National Institutes of Health | | Jinfang Zhu |
| National Institutes of Health | P30AR048335 | Wayne M Yokoyama |
| National Institutes of Health | T32 CA009547 | Dorothy K Sojka |

The funders had no role in study design, data collection and interpretation, or the decision to submit the work for publication.

### Author contributions

DKS, Conception and design, Acquisition of data, Analysis and interpretation of data, Drafting or revising the article; BP-D, Conception and design, Analysis and interpretation of data; LY, MAP-W, JY, Conception and design, Acquisition of data; MNA, YI, CZ, Conception and design, Acquisition of data, Analysis and interpretation of data; JMC, Conception and design, Drafting or revising the article; PBR, ZT, Conception and design, Drafting or revising the article, Contributed unpublished essential data or reagents; JKR, Acquisition of data, Analysis and interpretation of data; JZ, WMY, Conception and design, Analysis and interpretation of data, Drafting or revising the article

### Ethics

Animal experimentation: All mice were housed in a pathogen-free facility and all procedures were performed in accordance with the animal protocol (#20130049) approved by the Washington University School of Medicine (WUSM) Animal Studies Committee or the NIAID Animal Care and Use Committee.

## Additional files

### Supplementary files

• Supplementary file 1. Bone marrow DX5⁻ NK cell signature genes. These genes were used in the expression analysis shown in *Figure 2D*.

### Major dataset

The following dataset was generated:

| Author(s) | Year | Dataset title | Dataset ID and/or URL | Database, license, and accessibility information |
|---|---|---|---|---|
| Artyomov Maxim N | 2014 | Tissue-Resident Natural Killer (NK) Cells Are Cell Lineages Distinct From Thymic and Conventional Splenic NK Cells | http://www.ncbi.nlm.nih.gov/geo/query/acc.cgi?acc=GSE52047 | Publicly available at GEO (http://www.ncbi.nlm.nih.gov/geo/). |

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
