## [Decision Letter]

Thank you for sending your work entitled “Tissue-Resident Natural Killer (NK) Cells Are Cell Lineages Distinct From Thymic and Conventional Splenic NK Cells” for consideration at *eLife*. Your article has been favorably evaluated by a Senior editor and 3 reviewers, one of whom is a member of our Board of Reviewing Editors, and one of whom, Eric Vivier, has agreed to reveal his identity.

The Reviewing editor and the other reviewers discussed their comments before we reached this decision, and the Reviewing editor has assembled the following comments to help you prepare a revised submission.

Yokoyama and colleagues describe characteristics of tissue-resident NK cells (trNK) in the liver, uterus and skin. Based on phenotypic and functional analysis as well as dependency on several transcription factors, the authors conclude that at least 4 lineages of NK cells can be defined: conventional NK cells that circulate in the blood (cNK), thymic NK cells, and two types of trNK cells (liver/skin versus uterus). NK cell diversity is a subject of interest as NK cell function may vary in different tissues depending on environmental cues. This knowledge is important in order to understand how NK cells act during immune responses and also if they have roles in tissue homeostasis. This manuscript is quite interesting and for the most part the experiments are well performed and analyzed. The following points should be addressed in a revised version:

1) The experiments and findings from this study overlap, to some degree, with a previous report from the same group, Peng et. al. 2013. In the published study, they identified the unique and stable CD49a^+^DX5^-^ liver NK population (the trNK) and showed that it is phenotypically and functionally distinct from cNK, including differential hapten memory. The present study expands on the phenotypic and functional differences between trNK and cNK, but its main point, according to the title, is that the trNKs represent a distinct lineage. However, this conclusion is based largely on differential dependencies on transcription factors. And the only really convincing case is that of NFIL3 – the other dependencies could reflect effects on survival, niche residency, etc. The manuscript would be bolstered by concordant data from more classical ways to weigh lineage sharing, for example, genetic tagging experiments or precursor–product assessments. For example, concerning the latter, and given recent data on tissue macrophages, are both trNK and cNK derived from bone-marrow HSCs? A relatively easy experiment to do.

2) Relatedly, the authors study trNK cells in mice deficient for several transcription factors (Gata3, Tbet, Nfil3) to understand which are required for trNK cell development. While it is clear that trNK cells are present in the different KO strains, it would be useful to have absolute numbers for these studies (Gata3) and some additional phenotypic and functional confirmation that the trNK cells that are present are normal. This would be especially important for the *Nfil3* KO mice.

3) The RNAseq analysis is weak in a number of areas, or should at least be presented in a different manner. Figure 2 presents the top 41 differentially expressed genes as a heat-map, which disregards the other thousands of genes that might be similar or dissimilar. It's important to see the larger landscape. Figure 2 gets closer to this but the replicates do not look similar to each other and seem to not agree with the authors' conclusions. Dx5^+^ Liver 1 and Dx5^+^ Spleen 1 look more similar to CD49a^+^ Liver 1 and 2 than to their replicate counterparts. So, in the end, these data don't really add much. In addition, there is only limited information as to what constitutes the different NK cell signatures in cNK versus trNK cells. It would be helpful if the authors could provide the DX5^+^ and CD49a^+^ specific signatures (in BM and liver).

4) The authors should show the data relative to the presence of skin and uterus trNK cells in nude and GATA-3 deficient mice.

5) It has been recently reported in a meeting that liver trNK cells can produce IL-2. The authors should document whether they also detect IL-2 secretion by liver, skin and uterus trNK cells.

6) The antibody used to study NFIL3 protein expression (Figure 5) is not indicated. Staining specificity should be documented using the KO strains (for Nfil3, Tbet).

7) Concerning the unique identification of ILC in solid organs (Discussion), previous reports have described ILC in the skin (ILC2) and IL-22-producing ILC in the uterus (ILC3). The authors should revise this part of the Discussion.

---

## [Author Response]

*1) The manuscript would be bolstered by concordant data from more classical ways to weigh lineage sharing, e.g., genetic tagging experiments or precursor–product assessments. For example, concerning the latter, and given recent data on tissue macrophages, are both trNK and cNK derived from bone-marrow HSCs? A relatively easy experiment to do*.

In the Peng et al paper (JCI, 2013), our collaborators showed that the liver DX5−trNK cells are not completely reconstituted upon transfer of bone marrow (BM) cells to irradiated mice. We have similar results in our lab. The partial reconstitution could be due to several reasons. First, there may be a renewing precursor cell in the liver that generates the population, as suggested by experiments in the Peng paper where liver lineage^−^ cells could give rise to CD49a^+^DX5^-^ trNK cells in the liver. Second, precursor cells from fetal liver may seed the organ early on in life and adult BM HSCs only partially retain this property. We agree that the origin of the trNK cells in the various organs is very interesting and we had already mentioned these points in the Discussion, but based on the published and current results, further dissection of this issue appears to require an undertaking that exceeds the scope of the current manuscript.

*2) While it is clear that trNK cells are present in the different KO strains, it would be useful to have absolute numbers for these studies (Gata3)*…

Since GATA3 is important for the generation of thymic CD127^+^ NK cells, *Gata3* ko mice do not have thymic NK cells, allowing us to study these mice to determine if the trNK cells are related to thymic NK cells. However, *Gata3* ko mice are embryonic lethal (Nature 384:474, 1996), such that *Gata3* floxed mice need to be studied, as we did in our original submission. Since the manuscript has been submitted, however, our collaborators at the NIH have informed us that the *Gata3*^fl/fl^ x Vav-Cre mouse line is currently breeding very poorly and has yielded very few viable adults for analysis. While we have waited for some time to obtain these mice they have not been forthcoming so we decided to take a different route. Note however that the original experiments included several replicates so we are confident of the original GATA3 data that will remain in the manuscript.

We performed additional experiments to address the concern regarding the thymus and trNK cells by two other means to determine if the thymic NK cells are somehow related to the trNK cells. 1) We stained the trNK cells for CD127^+^ expression in the liver, skin, uterus and compared the expression to the thymic NK cells as well as the cNK cells in each of the organs (new Figure 7). In contrast to thymic NK cells, which express high levels of CD127, we found no CD127^+^ expression on any of the trNK cells tested. 2) Additionally the analysis of the *Foxn*^−/-^ (nu/nu) mice was extended to include absolute numbers in the spleen, liver, uterus and skin (new Figures 1 and 7). The *Foxn*^−/-^ (athymic) mice had no defect in the absolute number of either the trNK cells in the liver, uterus and skin or the conventional NK cells. Thus, thymic NK cells, which require an intact thymus for development, are distinct from the trNK cells.

*…and some additional phenotypic and functional confirmation that the trNK cells that are present are normal. This would be especially important for the Nfil3 KO mice*.

As suggested by the reviewers, additional data are now included in new Figure 5, G, and H and I. Tissue-resident NK cells from the NFIL3^-/-^ mice phenotypically show a similar activated status as trNK cells isolated from WT mice at steady state (Figure 5) and a higher CD27 expression similar to WT (Figure 5). They also do not express Eomes, like WT trNK cells (Figure 5). Finally, the NFIL3-deficient trNK cells produce cytokines like their WT counterparts (Figure 5). In order to include absolute numbers of trNK cells in the *Nfil3*^-/-^ mice and the cNK cells in the *Tbx21*^-/-^mice, a new figure (Figure 8) was added which contains some of the data from original Figure 6. Thus, the trNK cells in NFIL3-deficient mice appear phenotypically similar to trNK cells in WT mice.

*3) The RNAseq analysis is weak in a number of areas, or should at least be presented in a different manner.*
Figure 2
*presents the top 41 differentially expressed genes as a heat-map, which disregards the other thousands of genes that might be similar or dissimilar. It's important to see the larger landscape*.

As advised, we replaced Figure 2 with a figure that displays the entire data set of the hierarchical clustering analysis. Because we lost resolution, we highlighted specific genes as shown in the previous Figure 2 by moving it to Figure 2—figure supplement 1.

Figure 2
*gets closer to this but the replicates do not look similar to each other and seem to not agree with the authors' conclusions. Dx5*^*+*^
*Liver 1 and Dx5*^*+*^
*Spleen 1 look more similar to CD49a*^*+*^
*Liver 1 and 2 than to their replicate counterparts. So, in the end, these data don't really add much. In addition, there is only limited information as to what constitutes the different NK cell signatures in cNK versus trNK cells. It would be helpful if the authors could provide the DX5*^*+*^
*and CD49a*^*+*^
*specific signatures (in BM and liver)*.

While there is some variability in the replicates of the conventional NK cells in the spleen and the liver, what we aimed to show in the Figure is that the transcriptional profile of the “immature” trNK cells in the liver does not resemble the transcriptional profile of the “immature” BM NK cells. These data suggest that the trNK cells are not merely “immature” NK cells from the BM but could represent a distinct population of NK cells, as we went on to show by other means, such as by differential transcription factor-dependence. We have replaced the original figure with what we think is a clearer visual representation of the data and moved original Figure 2 to supplementary data (Figure 2—figure supplement 2). We provide the BM DX5^-^ gene signature in [Supplementary-material SD1-data].

*4) The authors should show the data relative to the presence of skin and uterus trNK cells in nude and GATA-3 deficient mice*.

As mentioned in point 2 above, the *Gata3*^fl/fl^-Vav-Cre mice are currently unavailable from our collaborators for further analysis due to difficulty in breeding. However, we analyzed the *Foxn*^−/-^ (nu/nu) mice and now provide the absolute numbers of both the trNK cells in the liver, spleen, uterus and skin as well as the cNK cells in these organs (new Figures 1 and 7). The trNK cells can be found in the uterus as well as the skin of the *Foxn*^−/-^ mice indicating that the trNK cells do not need a thymus to develop.

*5) It has been recently reported in a meeting that liver trNK cells can produce IL-2. The authors should document whether they also detect IL-2 secretion by liver, skin and uterus trNK cells*.

As requested, IL-2 secretion was measured and now can be found in a new Figure 7. Tissue-resident NK cells in the liver, uterus, and skin produce IL-2 whereas cNK cells do not, providing another example of the similarity between the trNK cells as distinct from cNK cells.

*6) The antibody used to study NFIL3 protein expression (*Figure 5*) is not indicated. Staining specificity should be documented using the KO strains (for Nfil3, Tbet)*.

Upon further investigation, our original Figure 5 appears to be misleading. What we wanted to emphasize is that there is little difference in NFIL3 expression by trNK and cNK cells, consistent with the transcript analysis (Figure 5). However, it appears that there is high non-specific staining and the baseline expression of NFIL3 protein is actually lower in NK cells. To avoid misleading the reader, we have decided to take out the staining with anti-NFIL3. This does not change the original point or the ultimate message of the paper. We thank the reviewers for bringing this to our attention.

The Tbet antibody has been used previously in prior publications, including a paper by the first author (Sojka, PNAS 2011), which we now reference.

*7) Concerning the unique identification of ILC in solid organs (Discussion), previous reports have described ILC in the skin (ILC2) and IL-22-producing ILC in the uterus (ILC3). The authors should revise this part of the Discussion*.

We revised the Discussion to include discussion points about the ILC2’s in the skin and ILC3’s in the uterus.